# Design and Multi-Parameter Optimization of a Combined Chinese Milk Vetch (*Astragalus sinicus* L.) Seed Harvester

Zhaoyan You [1], Xuemei Gao [1], Jianchun Yan [1], Hai Wei [1], Huichang Wu [1,*], Tieguang He [2] and Ji Wu [3]

1   Nanjing Institute of Agricultural Mechanization, Ministry of Agriculture and Rural Affairs, Nanjing 210014, China
2   Agricultural Resource and Environment Research Institute, Guangxi Academy of Agricultural Sciences, Nanning 530007, China
3   Soil and Fertilizer Institute, Anhui Academy of Agricultural Sciences, Hefei 230031, China
*   Correspondence: wuhuichang@caas.cn

**Abstract:** In order to solve problems such as poor applicability of headers, weak separation ability of threshing mechanisms and poor impurity-removal ability of cleaning devices in the existing seed harvest methods of Chinese milk vetch (*Astragalus sinicus* L.), a combined Chinese milk vetch seed harvester was designed in this paper. The parameters of the key components, such as the flexible anti pod-dropping seedling-lifting header, the longitudinal rod-teeth-type threshing device and the air-sieve-type layered impurity-controlled cleaning device, were designed and optimized. Aiming at reducing seed loss rate, breakage rate and impurity rate of Chinese milk vetch during the mechanical harvesting process, through multi-parameter optimization, the best combination of working parameters was obtained: machine forward speed was 3 km·h$^{-1}$, rotation speed of the threshing drum was 550 r·min$^{-1}$, rotation speed of the cleaning fan was 990 r·min$^{-1}$ and the scale sieve's opening was 35 mm. Field tests were performed under these parameters, and the results showed that the seed loss rate of Chinese milk vetch was 2.35%, the breakage rate was 0.22% and the impurity rate was 0.51%, which were better than the technical requirements of loss rate and breakage rate less than 5% and impurity rate less than 3% specified in relevant standards. The research results can solve the shortage problem of efficient seed harvest equipment in large-scale planting areas of Chinese milk vetch, and will further help to carry out seed harvest experiments on different varieties of Chinese milk vetch and other green manure varieties in paddy fields.

**Keywords:** Chinese milk vetch; green manure; seed harvest; parameter optimization; orthogonal test

## 1. Introduction

China is the country of origin of Chinese milk vetch (*Astragalus sinicus* L.), and also the country with the earliest utilization and cultivation of Chinese milk vetch and the largest planting area in the world. Chinese milk vetch, also known as grass seed, zi yunying and so on, is one of the main winter green manure crops in paddy fields of central and southern China [1–3]. In the 1970s, the planting area of Chinese milk vetch once exceeded 6.7 million square kilometers, mainly distributed in Sichuan, Hubei, Hunan, Jiangxi, Anhui, Jiangsu, Zhejiang and other provinces to the south of the Yangtze River. It has a strong ability of nitrogen fixation and high utilization efficiency, which can improve soil fertility and protect the ecological environment [4–6]. Currently, the harvest methods of Chinese milk vetch green manure seeds mainly include artificial harvest and mechanical harvest. Artificial harvest is time-consuming and laborious, and the yield of reserved seeds for planting is low. Generally, the seed yield of Chinese milk vetch in a paddy field is 300~650 kg·hm$^{-2}$. There are two common methods of mechanical harvest: the first one is segmented harvest, which uses a rice, wheat, rape or bean swather to harvest Chinese milk vetch, and then, through natural drying, a thresher is used for threshing afterwards; the second one is combine harvest, done by adjusting parameters and changing working components of a traditional

grain harvester or rapeseed combine harvester to complete seed harvesting of Chinese milk vetch [7]. The segmented harvesting process is cumbersome and inefficient, and combine harvesting is of high efficiency, which is the development trend of Chinese milk vetch seed harvesting, however, the harvest quality of existing Chinese milk vetch green manure seed combine harvesters is affected by the unreasonable structure configuration of headers, weak separation ability of threshing mechanisms and poor impurity-removal function of cleaning devices. The loss rate of machine harvesting in field tests is in the range from 31.5% to 32.1% [8], which seriously affects the scale promotion and application of Chinese milk vetch.

At present, existing research on seed combine harvesters, both at home and abroad, mainly focuses on food crops such as rice, wheat, maize, as well as commercial crops such as rapeseed, soybean and flax. Wang et al. [9] designed a cutting table to be a stepless speed-adjustable telescopic structure to harvest rapeseed, and the threshing device was designed to be a longitudinal-axis drum with the same diameter and different speed. Wang et al. [10] developed a cleaning device with segmented vibrating screens, whose holes were round so that the cleaning rate and loss rate of maize grain could meet the requirements of national standards for maize grain harvesters. Zhang et al. [11] used the Plackett–Burman test method to study the impacts of vibration screen amplitude, crank revolving speed, fan revolving speed and fan dip angle on cleaning loss ratio and impurity percentage of rapeseed based on a two-roller and air-screen field mobile harvest testbed. Jin et al. [12] used a series of field trials to explore the influence of nine key working parameters on the quality of soybean harvesting operations, and figured out the optimal combination of parameters systematically. Shi et al. [13] designed a track combine harvester for hilly mountain flax, which included a crawler-type walking system, a low-damage header to prevent winding, a transverse-flow beater with the grain rod and rod teeth with small taper and narrow-grid concave plates, but the impurity rate was relatively high. Bruce et al. [14] studied the effect of threshing rotor speed and concave clearance on the threshing performance of shattering rape pods. Mekonnen et al. [15] presented the effect of a cross-flow opening on the distribution of flow along the width of a forward curved, wide centrifugal fan with two parallel outlets, and Computational Fluid Dynamics (CFD) was utilized to study the effect of the addition of a cross-flow opening on the performance of the fan using three fans of similar geometries but different in their cross-flow opening. Shreekant et al. [16] studied the effects of mechanical damage of soybean seeds during vertical bucket lifting, cleaning and grading on seed germination and seed activity.

Existing studies have provided some research on the seed combine harvester, however, there have been much fewer reports on the combined seed harvest of Chinese milk vetch. Therefore, in this paper, the Chinese milk vetch (*Astragalus sinicus* L.) seed combine harvester was developed based on the "World Group 4LZ-5.0E Ryzen Grain Harvester", the structure and movement parameters of the key components, such as the flexible anti pod-dropping seedling-lifting header, the longitudinal rod-teeth-type threshing device and the air-sieve-type layered impurity-controlled cleaning device were newly designed and optimized according to the harvest characteristics of Chinese milk vetch. The factors that had great influence on the seed harvest quality of Chinese milk vetch were chosen to be machine forward speed, rotation speed of the threshing drum, rotation speed of the fan and the scale sieve's opening. A four-factor and three-level response surface test was carried out, the effects of various factors on the evaluation indexes of seed harvest quality of Chinese milk vetch were explored and the ideal combination of parameters was obtained as well. On this basis, a combined Chinese milk vetch green manure seed harvester was developed, which could complete the work of lifting, dividing, cutting, conveying, threshing, cleaning and unloading of Chinese milk vetch at one time. The performance of the machine was verified by field test, which provides reference for the development and application of mechanized seed harvesting technology and equipment for different Chinese milk vetch varieties and other green manures.

## 2. Materials and Methods

Chinese milk vetch (*Astragalus sinicus* L.) is a legume plant. The growth states of Chinese milk vetch at different periods are shown in Figure 1. It is an expert at using rhizobia to fix nitrogen, as shown in Figure 1a. A large amount of nitrogen, required by rhizobia in the growth process, is converted into ammonia nitrogen through the nodules, and what it returns to the rhizobia is the carbohydrates they need for survival. In this way, every part of the whole unit enjoys mutual benefit and peaceful symbiosis. Fruit pods of Chinese milk vetch at harvest time are shown in Figure 1b. When the average black pod rate of Chinese milk vetch reaches more than 75%, it should be harvested by machine in time.

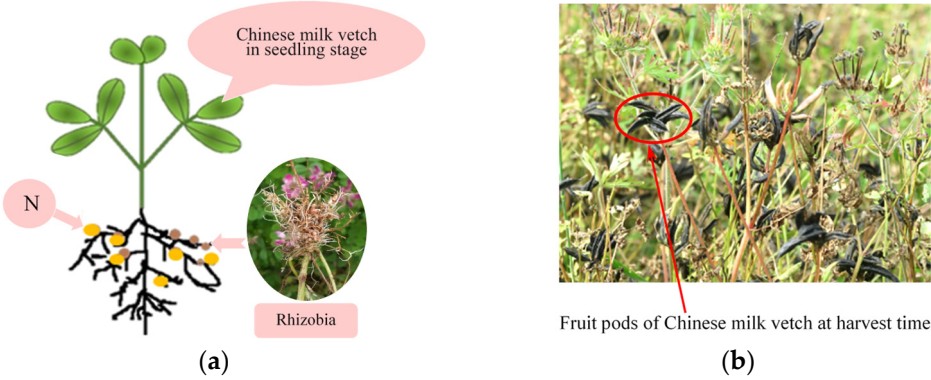

(**a**) (**b**)

**Figure 1.** Growth states of Chinese milk vetch at different periods. (**a**) Nitrogen-fixation effect picture of Chinese milk vetch in seedling stage. (**b**) Chinese milk vetch during harvest.

### 2.1. The Overall Structure and Technical Parameters

The structure and operation process of the developed Chinese milk vetch seed combine harvester is shown in Figure 2. The structure of the Chinese milk vetch green manure seed harvester is mainly composed of a reel, divider, grain lifter, conveying device, crawler chassis, cleaning device, seed bin, threshing device and discharge device. The key technical parameters of the Chinese milk vetch green manure seed combine harvester are shown in Table 1.

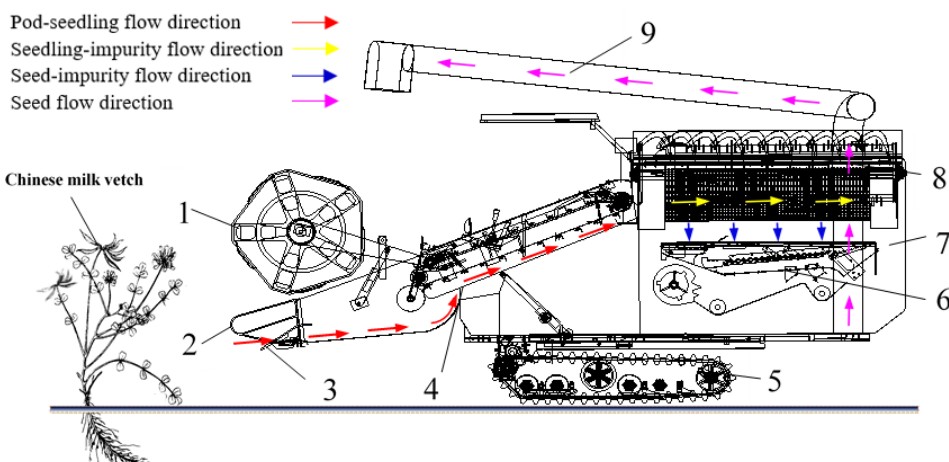

**Figure 2.** Structure and operation process diagram of the Chinese milk vetch green manure seed combine harvester: (1) reel; (2) divider; (3) grain lifter; (4) conveying device; (5) crawler chassis; (6) cleaning device; (7) seed bin; (8) threshing device; (9) discharge device.

**Table 1.** Key technical parameters of the Chinese milk vetch seed combine harvester.

| Parameters | Design Values |
|---|---|
| Machine size (L × W × H)/(mm) | 5100 × 2890 × 2700 |
| Overall weight/kg | 2800 |
| Matched power/kW | 62 |
| Operation width/mm | 2200 |
| Maximum feeding quantity/(kg·s$^{-1}$) | 5.0 |
| Work efficiency/(hm$^2$·h$^{-1}$) | 0.53~0.87 |
| Loss rate/% | ≤5 |
| Breakage rate/% | ≤2 |
| Impurity rate/% | ≤3 |

*2.2. Working Principle*

When the Chinese milk vetch seed combine harvester works in the field, the pods and stems of Chinese milk vetch enter the longitudinal rod-teeth-type threshing device under the combined action of the reel, the flexible anti pod-dropping seedling-lifting header and the conveying device. At this time, the pods of Chinese milk vetch are threshed under the rotating strike of the rod teeth and extruded by the concave sieve. Then, the threshed mixture falls into the sieve surface of the layered impurity-controlled cleaning device and the air cleaning of threshed materials under the action of a centrifugal fan is completed. The Chinese milk vetch seeds fall along the gaps between the sieving slices of the scale sieve, and fall into the horizontal seed auger through the round-hole screen. The seed auger directly transports them to the seed bin, and a small part of the unthreshed materials are not blown out of the machine; instead, seeds fall into the horizontal residual auger, and the residual auger transports this part of materials back to the threshing device for secondary threshing and cleaning so as to reduce the loss rate and impurity rate of seeds and complete the harvest operation of Chinese milk vetch seeds.

**3. Design of Critical Components**

*3.1. Flexible Anti Pod-Dropping Seedling-Lifting Header*

Due to the influence of weather, the harvest of Chinese milk vetch is usually creeped when it becomes mature. In order to reduce the seed loss rate and improve the harvest quality of Chinese milk vetch, the flexible anti pod-dropping seedling-lifting header was designed, as shown in Figure 3a. It is mainly composed of a reel, a divider, four lifters, cutter components, a depth-control equipment, a feeding auger and a side cutter, which can complete the operations of dividing, supporting, poking, cutting, guiding and conveying at one time.

The special lifter for the Chinese milk vetch seed harvester, as shown in Figure 3b, is mainly composed of a seedling-lifted rod head, spring rod connecting plate, spring rod and mounting plate. The seedling-lifted rod head was kept at an angle of 32~38° with the ground, and the stems and pods of Chinese milk vetch were guided to the rear by the front seedling-lifted rod head to lift the creeping or lodging Chinese milk vetch to be harvested, so as to avoid the cutter directly contacting the growth segment of Chinese milk vetch pods, and effectively reduce the header loss rate of the Chinese milk vetch seed combine harvester.

The cutter components, as shown in Figure 3c, are composed of a blade guard, cutting blade, left pushed saw-teeth, right pushed saw-teeth, transmitted connecting rod, etc. There were 26 blade guards in total, the type of blade guard was type II double finger, there was a tongue guard at the top of the tip to form a double support for the stems of Chinese milk vetch and the type of cutting blade was standard type II with a blade thickness of 2~3 mm and 6~7 teeth per centimeter of blade length.

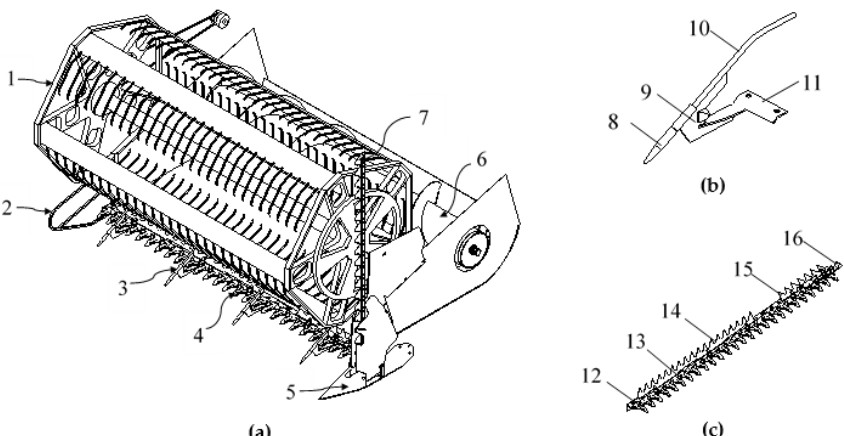

**Figure 3.** Schematic diagram of the flexible anti pod-dropping seedling-lifting header and its key parts: (1) reel; (2) divider; (3) lifter; (4) cutter components; (5) depth-control equipment; (6) feeding auger; (7) side cutter; (8) seedling-lifted rod head; (9) connecting plate of spring rod; (10) spring rod; (11) mounting plate; (12) blade guard; (13) cutting blade; (14) left pushed saw-teeth; (15) right pushed saw-teeth; (16) transmitted connecting rod. (**a**) Schematic diagram of the header. (**b**) Lifter for harvesting Chinese milk vetch. (**c**) Cutter components.

### 3.2. Longitudinal Rod-Teeth-Type Threshing Drum

In order to enhance the threshing performance of the threshing device, a longitudinal rod-teeth-type threshing drum was designed (shown in Figure 4), mainly consisting of a top cover, guide plate, spoke disc, feeding section auger, front end cap, main axle, mounting tube, rod teeth, concave sieve, bearing block, rear end cap, etc. The threshing roller is divided into the feeding section, threshing section and grass-discharging section. The total length of the threshing section was 1550 mm, the length of the feeding section was about four-fifths of the total length of the threshing section [17], that is, 315 mm, the length of grass-discharging section was 148 mm, the top cover above the roller and the concave sieve formed a cylindrical threshing chamber and the inner wall of the top cover was equipped with spiral guide plates so that stems of Chinese milk vetch will move along the axis to the grass-discharging section in the joint action of threshing rod teeth and guide plates.

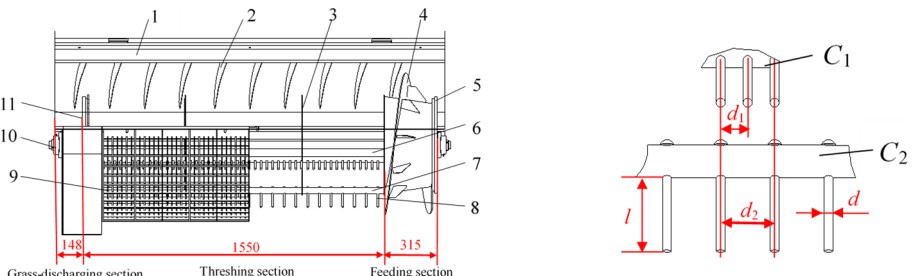

**Figure 4.** Schematic diagram of the longitudinal rod-teeth-type threshing device: (1) top cover; (2) deflector; (3) spoke disc; (4) auger of feeding section; (5) front end cap; (6) main axle; (7) mounting tube; (8) rod teeth; (9) concave sieve; (10) bearing block; (11) rear end cap.

The number of rod teeth on the roller is determined by the productivity of the threshing device [18],

$$z \geq (1 - \beta)q/0.6q_d \tag{1}$$

where, $z$ is the number of teeth on the roller; $q$ is the feed quantity of the thresher, set to 5 kg/s; $\beta$ refers to the weight proportion of crop pods in the feed materials, with the black pod harvest of Chinese milk vetch being 0.4; and $q_d$ is the threshing ability of each rod tooth (threshing ability is 0.025 kg·s$^{-1}$ for the combine harvester). In Equation (1), in the case $\beta = 0.4$, $q = 5$ kg·s$^{-1}$ and $q_d = 0.025$ kg·s$^{-1}$, the number of teeth on the roller is $z \geq 200$.

In order to improve the threshing performance, the teeth rows $C_1$ and $C_2$ with different rod-tooth spacing were installed alternately along the circumference of the drum, with the specific parameters designed as follows: rod-tooth spacing of teeth row $C_1$, $d_1$ was 30 mm; rod-tooth spacing of teeth row $C_2$, $d_2$ was 60 mm; working height of rod teeth, $l$, was 67 mm; diameter of rod teeth, $d$, was 10 mm. Through arrangement calculation, under the condition of ensuring the reliable working state of the threshing drum, the rod-teeth number of teeth row $C_1$ was 53, the rod-teeth number of teeth row $C_2$ was 27 and the total number of rod teeth on the roller was 240, which meets the productivity requirements of the Chinese milk vetch threshing device.

### 3.3. Air-Sieve-Type Layered Impurity-Controlled Cleaning Device

The cleaning structure of the Chinese milk vetch seed combine harvester was designed as an air-sieve-type layered impurity-controlled cleaning device. The structure diagram is shown in Figure 5. It is mainly composed of the centrifugal fan and the layered impurity-controlled cleaning sieve. In the air-sieve cleaning devices of the existing rice or wheat combine harvesters, most of the fans are single-channel centrifugal fans; when the structure and operation parameters are determined, the velocity and direction of air flow produced by single air-duct centrifugal fans cannot meet the different demands of the whole sieve in the process of material screening [19,20]. Referring to the structure characteristics of multi-air-duct centrifugal fans both at home and abroad, the number of blades of the centrifugal fan in the cleaning unit of Chinese milk vetch green manure seed combine harvester was designed as three blades, and the impeller diameter was 385 mm. Two adjustable inclined air flow dampers were installed at the air outlet of the fan, forming the upper, middle and lower air ducts, which can meet the cleaning requirements of the front, middle and rear of the layered impurity-controlled cleaning sieve.

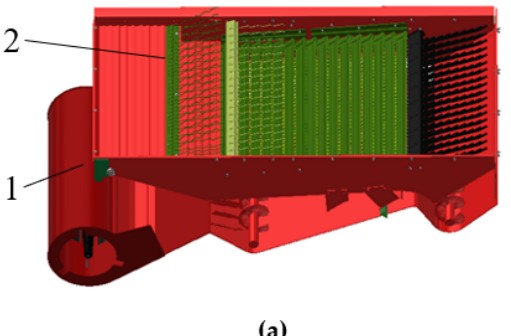 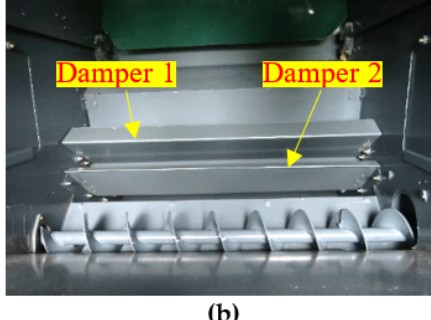

**(a)**          **(b)**

**Figure 5.** Three-dimensional and internal structure diagram of the air–sieve-type cleaning device: (1) centrifugal fan; (2) layered impurity-controlled sieve. (**a**) Three-dimensional diagram of the air–sieve-type cleaning device. (**b**) Internal structure diagram of the air–sieve-type cleaning device.

#### 3.3.1. Numerical Simulation of Internal Airflow Field in a Three-Duct Centrifugal Fan

In order to verify whether the flux and distribution of the internal airflow field of the designed three-duct centrifugal fan could meet the cleaning requirements of Chinese milk vetch seeds, the whole flow channel model of the three-duct centrifugal fan was established by using Inventor, and ICEM-CFD was adopted to divide the grid of the three-duct centrifugal fan. Considering the large volume of the channel model and the complexity of its internal flow, tetrahedral unstructured meshing was used to divide the flow channel model. The contact areas between the fan inlet and the impeller, as well as the contact areas between the impeller and the spiral case were all set as the interfaces, and the meshings at the interfaces were encrypted [21]. The meshed centrifugal fan is shown in Figure 6a, and the total number of model grids is 3,617,588.

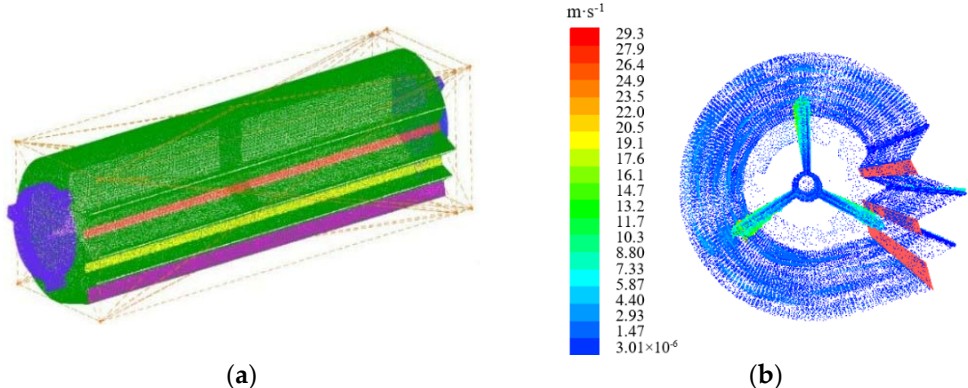

**Figure 6.** Mesh dividing and internal air velocity simulation of the three-channel centrifugal fan. (**a**) Mesh mode of centrifugal fan with three-channels. (**b**) Internal flow velocity vector diagram of the fan.

Then, the grid model was imported into Fluent 15.0 software, the standard *k-ε* turbulence model was used for numerical simulation and the SIMPLEC pressure-velocity coupling algorithm was used to calculate velocity and pressure of airflow in the fan. The spiral case of the fan was set as the static wall, and the meshing areas at the impeller of the centrifugal fan were set as the MRF rotation areas. Referring to the structure and working parameter design of existing centrifugal fans, rotation speed of the three-duct fan was set to be 1080 r·min$^{-1}$, the impeller diameter was 385 mm, the two inlet boundary conditions of the centrifugal fan were set as the pressure inlets with the given pressure of 220 Pa, the three outlet boundaries of the model were set as the pressure outlets with the given pressure of 0 Pa, the convergence residual was set as 0.001 and the iteration steps were set as 500 [22]. The internal airflow velocity vector diagram of the three-duct centrifugal fan is shown in Figure 6b. It can be seen from the figure that the airflow velocity at the upper and lower outlets was high, while the airflow velocity at the middle outlet was slightly lower. The airflow velocity at the upper outlet (regardless of friction resistance) ranged from 7.33 m·s$^{-1}$ to 29.3 m·s$^{-1}$, and the air volume at the outlet was about 0.55 kg·s$^{-1}$; the airflow velocity at the middle outlet ranged from 5.14 m·s$^{-1}$ to 24.9 m·s$^{-1}$, and the air volume was about 0.42 kg·s$^{-1}$; the wind speed of the lower outlet was between 6.6 m·s$^{-1}$ and 25.6 m·s$^{-1}$, and the air volume was about 0.57 kg·s$^{-1}$.

### 3.3.2. Field Test Verification of Centrifugal Fan Air Flow

The method of spot placement was adopted [23], according to the structural parameters of the three-duct centrifugal fan. The point at the lowest position on the volute area of the centrifugal fan and 50 mm away from the left inlet were taken as the origin points, the radial direction of seed-conveying auger was chosen as the *X* axis and the vertical direction of the sieve surface was chosen as the *Y* axis. The air velocity measurement space and the distribution of measuring points are shown in Figure 7.

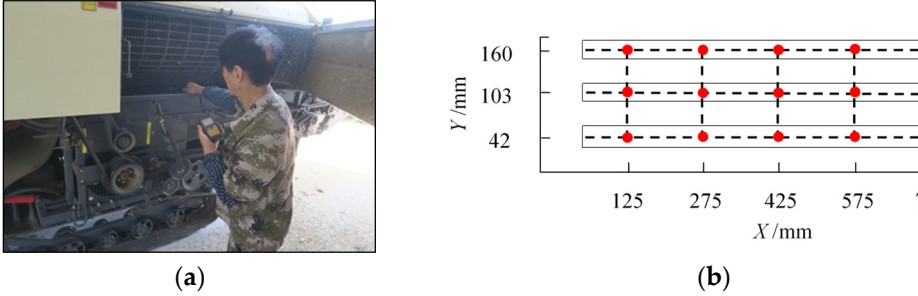

**Figure 7.** Air velocity measurement site and measurement points distribution. (**a**) Air velocity measurement. (**b**) Measurement points distribution.

The air velocity of each measuring point measured by AR856 digital anemometer is shown in Table 2. The sensitivity and features of the digital anemometer are as follows: airflow speed measurement ranged from 0.3 m·s$^{-1}$ to 45.0 m·s$^{-1}$, the measurement error was ±3% and the resolution ratio was 0.001 m·s$^{-1}$. Additionally, 3 planes of 42, 103 and 160 mm in the Y direction corresponded to 3 central measuring planes of air ducts, and each measuring plane was arranged with 5 measuring points at 125, 275, 425, 575 and 725 mm in the X direction. By comparing the results of numerical simulation and experimental study on airflow field, it can be seen that the airflow velocity distribution of each outlet was consistent. The horizontal airflows both at the upper and lower outlets were more uniform, which were conducive to the pre-cleaning of threshed materials and the discharge of impurities in the tail. The middle outlet presented the law of high airflow velocity in the middle and low airflow on both sides, which was conducive to the blowing and stratification of the threshed mixture during the falling process. In addition, the air velocity values were slightly different, the reason for which lied in that the numerical simulation process was completed without considering the gas compression and gas viscosity, and assuming that the whole flow channel was closed. However, in practical work, the cleaning air flow would be attenuated due to the existence of material groups [24].

**Table 2.** Airflow velocity of measuring points (m·s$^{-1}$).

| Y/mm | X/mm | | | | |
|---|---|---|---|---|---|
| | 125 | 275 | 425 | 575 | 725 |
| 42 | 9.7 | 11.1 | 10.7 | 8.8 | 9.2 |
| 103 | 6.2 | 5.7 | 10.1 | 6.6 | 6.4 |
| 160 | 9.2 | 8.6 | 10.3 | 7.7 | 8.1 |

### 3.3.3. Layered Impurity-Controlled Cleaning Sieve

There were many stalks in threshed materials of Chinese milk vetch, and the residual materials were mainly glumes, broken stalks and weed seeds. A schematic diagram of the layered impurity-controlled cleaning sieve and the scale sieve's opening is shown in Figure 8, where *y* is the opening of the scale sieve.

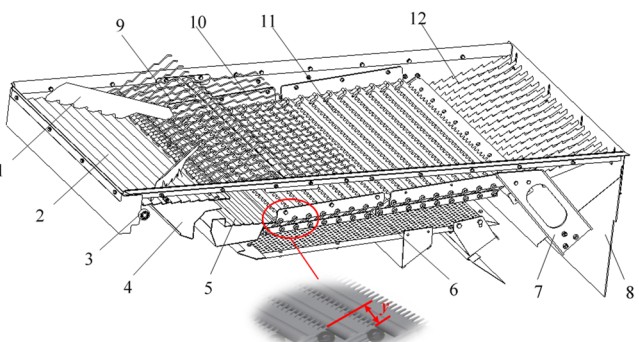

**Figure 8.** Schematic diagram of the layered impurity-controlled cleaning sieve and the scale sieve's opening: (1) baffle plate; (2) upper shaking plate; (3) bearing; (4) air deflector; (5) lower shaking plate; (6) round-hole sieve; (7) connecting plate; (8) rubber plate; (9) front finger sieve; (10) back finger sieve; (11) scale sieve; (12) saw-teeth sieve.

It was mainly composed of the baffle plate, upper shaking plate, lower shaking plate, air deflector, front finger sieve, back finger sieve, scale sieve, saw-teeth sieve and round-hole sieve. The front and back finger sieves separated the long stalks and residual spikes from the threshed materials under the sieve, the scale sieve was designed as a parallel four-bar linkage structure and the opening of the scale sieve was ranged between 35 mm and 45 mm, which could guide the air flow to blow away glume shell, residual spikes and broken straws, so as to ensure the high-efficiency screening of Chinese milk vetch

seeds. The round-hole sieve could further separate the Chinese milk vetch seeds from the mixture of grains and thin stems that were dropped from the scale sieve surface, while the saw-teeth sieve, mounted on the rear of the cleaning sieve, could discharge remaining long straws, residual spikes and broken straws out of the machine backward step by step or multistage. Structural parameters of the layered impurity-controlled cleaning sieve are shown in Table 3.

**Table 3.** Structural parameters of the layered impurity-controlled cleaning sieve.

| Parameters | Design Values |
| --- | --- |
| Upper shaking plate length/mm | 775 |
| Upper shaking plate width/mm | 334 |
| Upper shaking plate (spacing × height)/mm × mm | 28 × 6 |
| Front finger sieve length/mm | 740 |
| Front finger sieve width/mm | 225 |
| Back finger sieve length/mm | 680 |
| Back finger sieve width/mm | 224 |
| Lower shaking plate length/mm | 775 |
| Lower shaking plate width/mm | 203 |
| Lower shaking plate (spacing × height)/mm × mm | 30 × 4.5 |
| Scale sieve length/mm | 754 |
| Scale sieve width/mm | 715 |
| Saw-teeth sieve length/mm | 640 |
| Saw-teeth sieve width/mm | 328 |
| Round-hole sieve length/mm | 756 |
| Round-hole sieve width/mm | 756 |
| Diameter of round-hole sieve/mm | 6 |

## 4. Field Experiment and Result Analysis

### 4.1. Experiment Conditions

The field harvest experiment of Chinese milk vetch was conducted in Yijiang Town, Nanling County, Wuhu City, as shown in Figure 9. The experiment time was from 11–13 May 2020. The Chinese milk vetch variety used for the experiment was Wanzi No. 4, and its yield was 608.95 kg·hm$^{-2}$. The average height of Chinese milk vetch stems was 330.8 mm, the average height of the bottom pod was 118.5 mm, the average black pod rate was 80.68%, the natural seed loss was 3.31 g·m$^{-2}$ and one thousand seed weight was 3.3~3.5 g. The moisture content of harvested seeds was measured by a PM-8188-A moisture meter, and the moisture meter measurements ranged from 1% to 40%. The sample capacity was 240 mL, the temperature range was 0~40 °C, the measurement accuracy was 0.5% at the basis of the drying method and the water content of harvested seeds was measured three times, with an average value of 10.2%. The Chinese milk vetch plants grew well, and the average length and width of pods picked from 10 different Chinese milk vetch plants were 25.3 mm and 4.2 mm, respectively. The seeds in pods were kidney shaped, and about 3 mm in length.

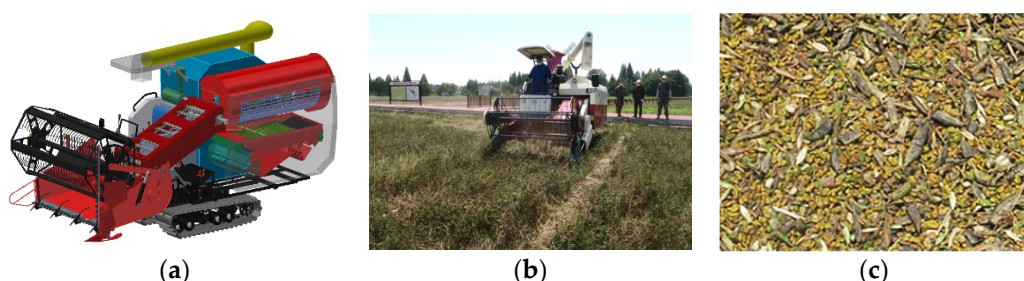

(**a**)　　　　　　　　　(**b**)　　　　　　　　　(**c**)

**Figure 9.** Combined Chinese milk vetch seed harvester and field test process. (**a**) Three-dimensional diagram of the combined Chinese milk vetch seed harvester. (**b**) Chinese milk vetch field harvest test. (**c**) Materials in the seed bin.

### 4.2. Experiment Indexes

At present, there is no evaluation standard for the operation of the Chinese milk vetch seed combine harvester. In order to investigate the operation quality of the Chinese milk vetch seed combine harvester, according to leguminosae or cruciferae green manure seed harvest standards such as JB/T 11912-2014, GB/T 5262-2008 and so on, seed loss rate, breakage rate and impurity rate are used as the evaluation indexes for the operation performance of the Chinese milk vetch seed combine harvester. Among them, the machine-harvested seed loss rate is determined by collecting all the seeds and pods, both those fallen in the sampling area and discharged out of the machine with Chinese milk vetch stems, and then removing the natural falling seeds. According to the quantity of harvested Chinese milk vetch seeds and the corresponding harvested area, the yield of Chinese milk vetch seeds per square meter is obtained. Impurities include long and short stalks, grass seeds, pebbles, etc. The broken seeds are incomplete cotyledons (including whole and half seeds), and transverse and broken grains. The specific calculation methods are shown in Equations (2)–(4):

$$Y_1 = \frac{M_{hl}}{M_{ha} + M_{hl}} \times 100\% \tag{2}$$

$$Y_2 = \frac{M_{ei} - M_{ps}}{M_{ei}} \times 100\% \tag{3}$$

$$Y_3 = \frac{M_{ss} - M_{ps}}{M_{ss}} \times 100\% \tag{4}$$

where, $Y_1$ is the loss rate, %; $Y_2$ is the breakage rate, %; $Y_3$ is the impurity rate, %; $M_{hl}$ is the seed loss quantity during harvest of Chinese milk vetch per square meter, $g \cdot m^{-2}$; $M_{ha}$ is the harvested seed quantity of Chinese milk vetch per square meter, $g \cdot m^{-2}$; $M_{ei}$ is the sample quality after removal of impurities, g; $M_{ps}$ is the sample quality after removal of impurities and broken seeds, g; $M_{ss}$ is the sample quality, g.

### 4.3. Experiment Scheme

According to the previous literature search, field experimental research and harvest experience [25,26], taking the loss rate, breakage rate and impurity rate of seeds as the evaluation bases of the operation performance of the Chinese milk vetch seed combine harvester, machine forward speed, rotation speed of the roller, rotation speed of the cleaning fan and the scale sieve's opening were selected as the test variables to carry out the experimental research. Box–Behnken central composite design theory was used to carry out a four-factor three-level quadratic regression response surface test with a total of 29 groups. The test site was about 120 m long and 50 m wide, and the test length of each group was 30 m. Samples were taken for three times in each group after the harvester ran stably.

Experiment factors and codes are shown in Table 4. Among them, $X_1 \sim X_4$ represented the variables of each factor. According to the design parameters of the seed combine harvester, the relationship between the biological characteristics of Chinese milk vetch during harvest and the threshing and cleaning parameters of the harvester, the value range of each factor was determined, the median value of machine forward speed was set as 4 $km \cdot h^{-1}$ and the low-speed value of 3 $km \cdot h^{-1}$ and high-speed value of 5 $km \cdot h^{-1}$ were selected accordingly. According to the threshing ability characteristics of Chinese milk vetch and referring to the rotation speed of the threshing drum for wheat, soybean and other crops, the middle value of rotation speed of the threshing drum was set as 675 $r \cdot min^{-1}$, the minimum rotation speed of the threshing drum was 550 $r \cdot min^{-1}$ and the maximum rotation speed was 800 $r \cdot min^{-1}$. According to the simulation results of the centrifugal fan in the early stage, meanwhile, referring to the test method and standard of seed suspension speed of grain, oil, forage grass, etc., [27], the minimum rotation speed of the cleaning fan was set as 900 $r \cdot min^{-1}$, the maximum rotation speed was 1260 $r \cdot min^{-1}$ and the intermediate value of 1080 $r \cdot min^{-1}$ was determined. Based on external dimensions of pods and seeds of Chinese milk vetch varieties, the designed opening range of the scale

sieve was from 35 mm to 45 mm, the middle value was 40 mm and the corresponding limit values of the opening range were taken as high- and low-level values.

**Table 4.** Experiment factors and codes.

| Factors | Codes | | |
|---|---|---|---|
| | **−1** | **0** | **1** |
| Machine forward speed/(km·h$^{-1}$) | 3 | 4 | 5 |
| Rotation speed of threshing drum/(r·min$^{-1}$) | 550 | 675 | 800 |
| Rotation speed of cleaning fan/(r·min$^{-1}$) | 900 | 1080 | 1260 |
| Scale sieve's opening/mm | 35 | 40 | 45 |

The adjustment methods of each parameter in the test were as follows: (1) Machine forward speed adjustment: The driver adjusted the infinitely variable gear to get different forward speeds. The forward speed was accurately detected by the speed sensor installed on the rear wheel and displayed on the instrument panel in the cab. (2) Rotation speed adjustments of the threshing drum and the cleaning fan: the V-belt with continuously variable transmission was adopted, and the moving plate of the belt wheel was adjusted by hydrostatic and manual operation so as to obtain different transmission ratios and different rotation speeds. (3) Scale sieve's opening adjustment: through the position rotation of the adjusting plate installed on the sieve frame, the scale's opening could be adjusted, and the specific scale's opening could be measured by ruler.

*4.4. Experiment Results and Analysis*

The experimental design, data processing and statistical analysis were conducted by the software Design-Expert V8.0.6.1. The four-factor and three-level responsive section experiment was carried out according to the Box–Behnken design method, which is to select 29 points, including 24 analysis factors and 5 zero estimation errors. The experimental design and response results are shown in Table 5, and the variance analysis of the regression model is conducted in Table 6.

**Table 5.** Experiment design and response values.

| No. | Codes | | | | Response Values | | |
|---|---|---|---|---|---|---|---|
| | $X_1$ | $X_2$ | $X_3$ | $X_4$ | $Y_1$ | $Y_2$ | $Y_3$ |
| 1 | −1 | 0 | 1 | 0 | 10.32 | 1.35 | 1.13 |
| 2 | 0 | 0 | −1 | 1 | 1.23 | 1.87 | 2.93 |
| 3 | 0 | 1 | 1 | 0 | 12.20 | 3.01 | 0.89 |
| 4 | 0 | 1 | −1 | 0 | 1.15 | 3.54 | 1.58 |
| 5 | 0 | 0 | 0 | 0 | 4.16 | 1.72 | 1.36 |
| 6 | 0 | 0 | 1 | 1 | 16.47 | 1.25 | 2.17 |
| 7 | 1 | 0 | 0 | 1 | 8.25 | 1.52 | 2.45 |
| 8 | 0 | −1 | 1 | 0 | 10.69 | 0.13 | 1.51 |
| 9 | 1 | 1 | 0 | 0 | 6.93 | 2.78 | 1.32 |
| 10 | 1 | 0 | 0 | −1 | 1.95 | 1.88 | 1.03 |
| 11 | 0 | −1 | −1 | 0 | 3.22 | 0.37 | 1.83 |
| 12 | 0 | 0 | 0 | 0 | 5.04 | 2.09 | 1.39 |
| 13 | 0 | −1 | 0 | 1 | 7.36 | 0.26 | 2.65 |
| 14 | 0 | 0 | 0 | 0 | 4.65 | 1.66 | 1.28 |
| 15 | 1 | 0 | −1 | 0 | 4.26 | 1.85 | 2.35 |
| 16 | −1 | 0 | −1 | 0 | 0.96 | 1.77 | 1.76 |
| 17 | 1 | 0 | 1 | 0 | 11.78 | 1.72 | 1.38 |
| 18 | 0 | 1 | 0 | −1 | 2.13 | 2.93 | 0.65 |
| 19 | 0 | −1 | 0 | −1 | 3.46 | 0.32 | 0.73 |
| 20 | 0 | 0 | 1 | −1 | 8.91 | 1.5 | 0.65 |
| 21 | 1 | −1 | 0 | 0 | 3.88 | 0.29 | 1.57 |
| 22 | −1 | −1 | 0 | 0 | 3.57 | 0.19 | 1.38 |
| 23 | 0 | 0 | 0 | 0 | 5.74 | 1.74 | 1.33 |
| 24 | −1 | 0 | 0 | 1 | 8.38 | 1.38 | 1.97 |
| 25 | −1 | 0 | 0 | −1 | 2.62 | 1.71 | 0.43 |
| 26 | 0 | 0 | 0 | 0 | 5.33 | 1.51 | 1.54 |
| 27 | −1 | 1 | 0 | 0 | 5.49 | 2.52 | 0.93 |
| 28 | 0 | 0 | −1 | −1 | 0.43 | 1.96 | 0.78 |
| 29 | 0 | 1 | 0 | 1 | 9.30 | 2.37 | 1.87 |

**Table 6.** Variance analysis of regression model.

| Indexes | Variance Source | Sum of Squares | F-Value | p-Value | Significance |
|---|---|---|---|---|---|
| $Y_1$ | $X_1$ | 2.72 | 2.08 | 0.1710 | |
| | $X_2$ | 2.10 | 1.61 | 0.2252 | |
| | $X_3$ | 291.26 | 223.29 | <0.0001 | *** |
| | $X_4$ | 82.64 | 63.35 | <0.0001 | *** |
| | Lack of fit | 16.78 | 4.52 | 0.0796 | |
| | Pure error | 1.48 | | | |
| $Y_2$ | $X_1$ | 0.10 | 2.40 | 0.1439 | |
| | $X_2$ | 20.25 | 464.38 | <0.0001 | *** |
| | $X_3$ | 0.48 | 11.01 | 0.0051 | *** |
| | $X_4$ | 0.23 | 5.20 | 0.0387 | ** |
| | Lack of fit | 0.43 | 0.94 | 0.5761 | |
| | Pure error | 0.18 | | | |
| $Y_3$ | $X_1$ | 0.52 | 27.05 | 0.0001 | *** |
| | $X_2$ | 0.49 | 25.56 | 0.0002 | *** |
| | $X_3$ | 1.02 | 53.03 | <0.0001 | *** |
| | $X_4$ | 7.95 | 413.18 | <0.0001 | *** |
| | Lack of fit | 0.23 | 2.39 | 0.2077 | |
| | Pure error | 0.039 | | | |

Note: ** indicates that the factors have a significant influence on the test index ($0.01 < p \leq 0.05$), *** indicates that the factors have a very significant influence on the test index ($p \leq 0.01$).

### 4.4.1. Model Establishment and Significance Verification

According to the sample data in Table 5, the regression models are established in Equations (5)–(7) as follows:

$$\begin{aligned} Y_1 \ = \ &4.98 + 0.48X_1 + 0.42X_2 + 4.93X_3 + 2.62X_4 + 0.28X_1X_2 \\ &-0.46X_1X_3 + 0.13X_1X_4 + 0.90X_2X_3 + 0.82X_2X_4 \\ &+1.69X_3X_4 + 0.018X_1^2 + 0.14X_2^2 + 1.67X_3^2 + 0.28X_4^2 \end{aligned} \tag{5}$$

$$\begin{aligned} Y_2 = \ &1.74 + 0.093X_1 + 1.30X_2 - 0.20X_3 - 0.14X_4 + 0.040X_1X_2 \\ &+0.072X_1X_3 - 0.0075X_1X_4 - 0.073X_2X_3 - 0.13X_2X_4 \\ &-0.04X_3X_4 - 0.1X_1^2 - 0.14X_2^2 + 0.065X_3^2 - 0.11X_4^2 \end{aligned} \tag{6}$$

$$\begin{aligned} Y_3 = \ &1.38 + 0.21X_1 - 0.20X_2 - 0.29X_3 + 0.81X_4 + 0.05X_1X_2 \\ &-0.085X_1X_3 - 0.03X_1X_4 - 0.093X_2X_3 - 0.17X_2X_4 \\ &-0.16X_3X_4 + 0.025X_1^2 - 0.074X_2^2 + 0.18X_3^2 + 0.10X_4^2 \end{aligned} \tag{7}$$

Through the analysis of Table 6, the regression models of seed loss rate, $Y_1$, breakage rate, $Y_2$, and impurity rate, $Y_3$, show that $p < 0.0001$. Moreover, the significance of each variable influencing the index in the regression equations is judged by the $F$ value. In other words, the smaller the probability value $P$ is, the higher the significance of the corresponding variable is. It can be seen from the $F$ value of each factor that rotation speed of the cleaning fan and the scale sieve's opening have a significant effect on seed loss rate. The descending order of influencing factors is rotation speed of the cleaning fan > scale sieve's opening > machine forward speed > rotation speed of the threshing drum. Meanwhile, rotation speed of the cleaning fan and threshing drum has a significant effect on seed breakage rate, and the descending order is rotation speed of the threshing drum > rotation speed of the cleaning fan > scale sieve's opening > machine forward speed. In addition, all the four factors have a significant effect on seed loss rate, and the descending order is scale sieve's opening > rotation speed of the cleaning fan > machine forward speed > rotation speed of the threshing drum. The determination coefficients, $R^2$, of $Y_1$, $Y_2$ and $Y_3$ are 0.9580, 0.9723 and 0.975, respectively, showing that the model error is small, and it is reasonable to analyze and predict seed loss rate, breakage rate and impurity rate of the Chinese milk vetch seed combine harvester.

### 4.4.2. Effect Analysis of Interaction Factors on Harvest Indexes

Based on the established optimization regression model, the influence of machine forward speed, rotation speed of the threshing drum, rotation speed of the cleaning fan and scale sieve's opening on the harvest quality index of the Chinese milk vetch seed combine harvester and the relationship among the factors were analyzed.

The response surfaces of seed loss rate are shown in Figure 10. In Figure 10a, the rotation speed of the threshing drum and scale sieve's opening were both zero, and the increase of machine forward speed had little influence on the seed loss rate when the rotation speed of the cleaning fan was fixed, and when machine forward speed was constant, seed loss rate gradually increased with the increase of the rotation speed of the cleaning fan. In Figure 10b, the machine forward speed was set at a low level, the rotation speed of the cleaning fan was set at a low level, when the value of threshing drum rotation speed was fixed, the seed loss rate increased with the increase of the scale sieve's opening, and when the scale sieve's opening was constant, the rotation speed of the threshing drum had little influence on the seed loss rate. Cause analysis: when the rotation speed of the cleaning fan and the scale sieve's opening increased, the air volume passing through the surface of the scale sieve in unit time would increase, and more seeds on the sieve surface would be blown away by the airflow; meanwhile, seeds that were prepared to pass the cleaning sieve would be blown out of the machine as well, resulting in the increase of seed loss rate.

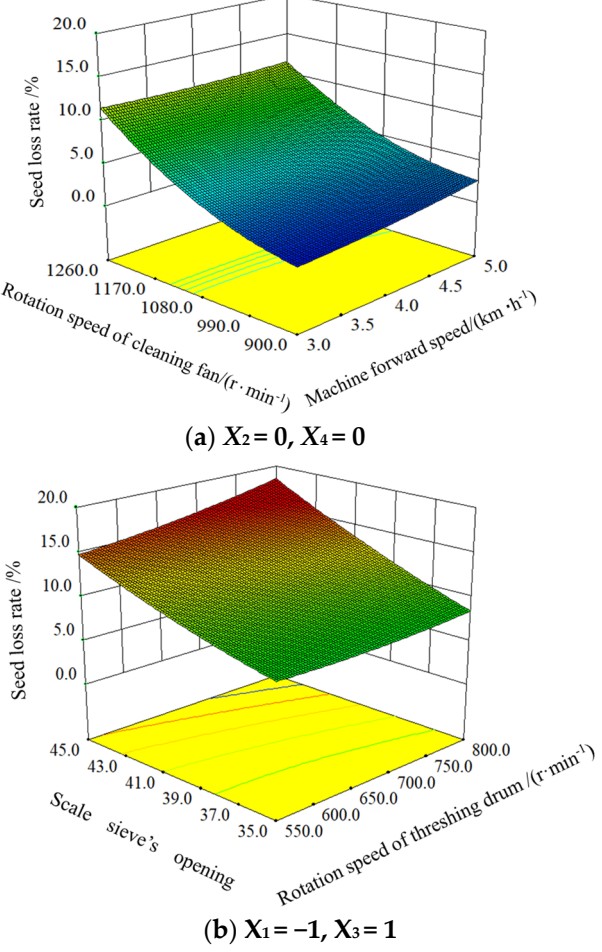

(**a**) $X_2 = 0$, $X_4 = 0$

(**b**) $X_1 = -1$, $X_3 = 1$

**Figure 10.** Response surfaces of seed loss rate.

The response surfaces of seed breakage rate are shown in Figure 11. Machine forward speed and the rotation speed of the cleaning fan were both zero, as shown in Figure 11a, when the scale sieve's opening was fixed, and the faster the rotation speed of the threshing

drum was, the higher the seed breakage rate was; when the rotation speed of threshing drum was constant, the scale sieve's opening had little influence on the seed loss rate. In Figure 11b, the scale sieve's opening and the rotation speed of the threshing drum were both zero, and when machine forward speed was fixed, the seed breakage rate decreased slightly with the increase of cleaning fan rotation speed; when rotation speed of the cleaning fan was constant, machine forward speed had little influence on the seed breakage rate. Cause analysis: the faster the rotation speed of the threshing drum, the harder the threshing element struck Chinese milk vetch seed, and the higher the possibility of seed breakage. At the beginning, with the increase in the cleaning fan rotation speed, the fan mainly blew away most of the impurities, and the broken seeds flowed into the seed bin through the round-hole screen, and seed breakage rate changed little. However, with further increase of the cleaning fan rotation speed, part of the damaged seeds were blown away, while some complete seeds on the surface of the sieve were also blown away by the fan. According to calculation Formula (3), the breakage rate increased correspondingly.

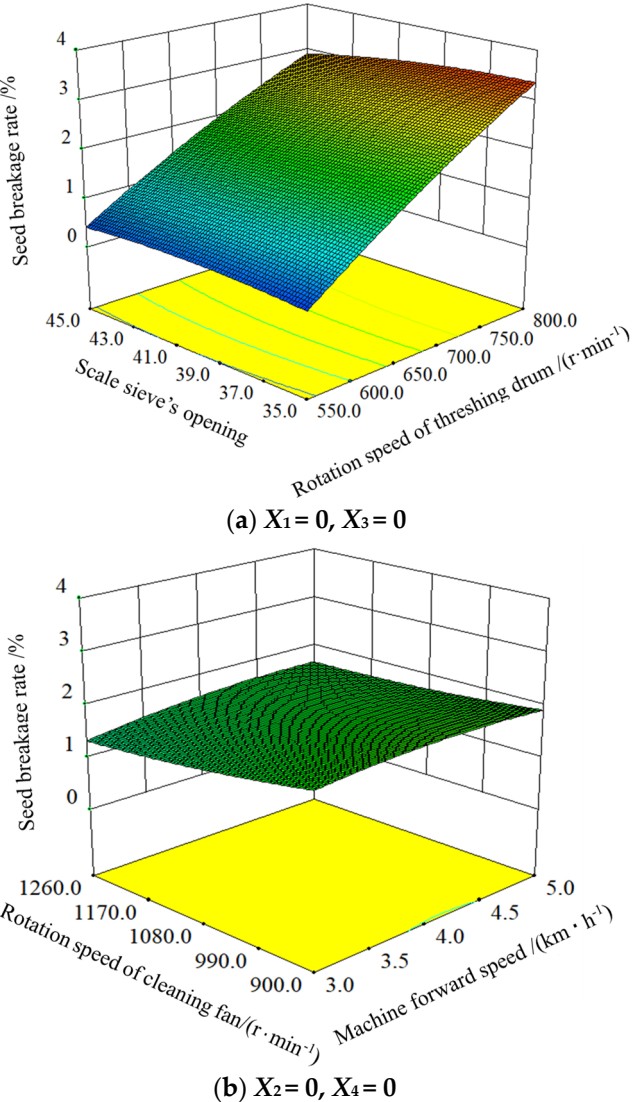

**(a)** $X_1 = 0, X_3 = 0$

**(b)** $X_2 = 0, X_4 = 0$

**Figure 11.** Response surfaces of seed breakage rate.

The response surfaces of seed impurity rate are shown in Figure 12. Rotation speed of the threshing drum and rotation speed of the cleaning fan were both zero, as shown in Figure 12a, when machine forward speed was constant, and the larger the scale sieve's opening was, the higher the seed impurity rate was; when the scale sieve's opening was fixed, with the increase of machine forward speed, the impurity content of seed also

increased slightly. In Figure 12b, the machine forward speed was set at a low level, and the scale sieve's opening was set at zero. When the rotation speed of the threshing drum was fixed, seed impurity rate decreased with the increase of cleaning fan rotation speed; when rotation speed of the cleaning fan was constant, the faster the rotation speed of threshing drum was, the smaller the seed impurity rate would be. Cause analysis: when the scale sieve's opening increased, most of the short stalks fell onto the round-hole screen from the gap of the sieve pieces, the number of stalks entering the round-hole sieve surface increased and the screening efficiency decreased, leading to the increase of impurity content. In addition, the lower the rotation speed of the cleaning fan was, the more surplus stalks would fall from the sieve pieces before they were blown away by the fan, resulting in incomplete cleaning separation and higher seed impurity content.

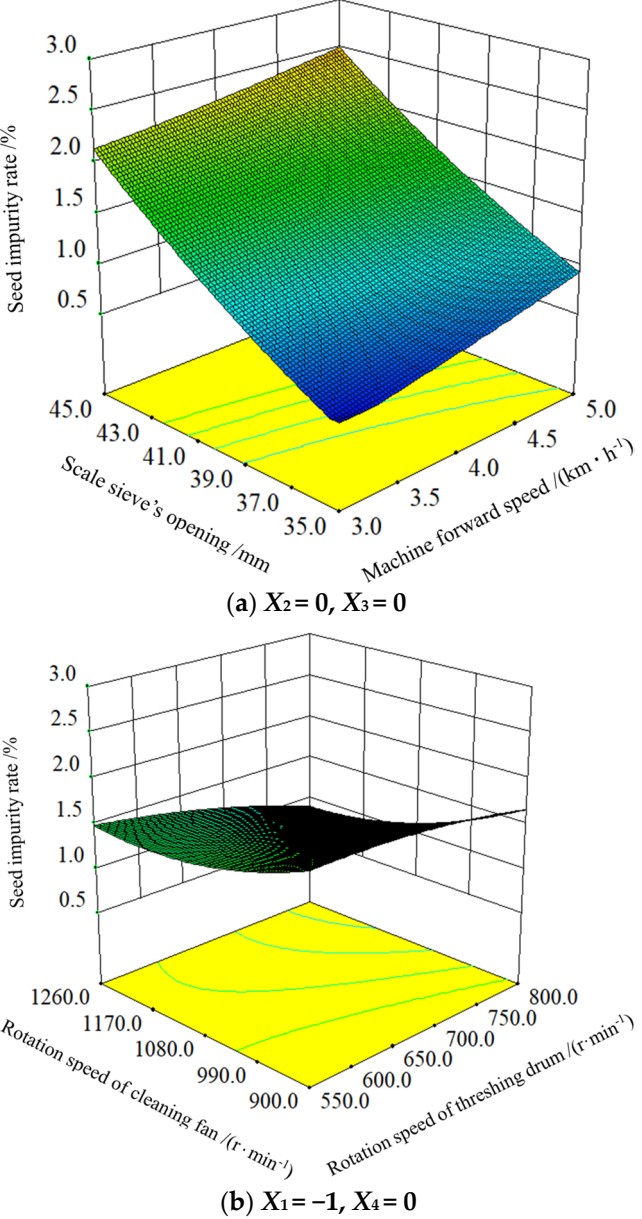

(**a**) $X_2 = 0$, $X_3 = 0$

(**b**) $X_1 = -1$, $X_4 = 0$

**Figure 12.** Response surfaces of seed impurity rate.

### 4.4.3. Parameter Optimization and Verification Test

In this paper, in order to meet requirements of lowest seed loss rate, lowest seed breakage rate and lowest seed impurity rate during the harvest of Chinese milk vetch, parameter optimization of the Chinese milk vetch green manure seed combine harvester

was carried out. Design-Expert data analysis software was used to optimize the established total factor quadratic regression model of the three indexes. Constraint conditions were established in Equation (8) as follows:

$$\begin{cases} \min Y_1 \\ \min Y_2 \\ \min Y_3 \\ \text{s.t.} \begin{cases} 3 \text{ km·h}^{-1} \le X_1 \le 5 \text{ km·h}^{-1} \\ 550 \text{ r·min}^{-1} \le X_2 \le 800 \text{ r·min}^{-1} \\ 900 \text{ r·min}^{-1} \le X_3 \le 1260 \text{ r·min}^{-1} \\ 35 \text{ mm} \le X_4 \le 45 \text{ mm} \end{cases} \end{cases} \tag{8}$$

The optimal parameter combination of the Chinese milk vetch seed combine harvester was obtained as follows: the machine forward speed was 3 km·h$^{-1}$, the rotation speed of the threshing drum was 553.45 r·min$^{-1}$, the rotation speed of the cleaning fan was 991.77 r·min$^{-1}$ and the scale sieve's opening was 35 mm, then, the seed loss rate, breakage rate and impurity rate predicted by the model were 2.40%, 0.19% and 0.48%, respectively.

In order to verify the accuracy of the above model, the validation test was carried out in Qingyijiang Town, Nanling County, Wuhu City, 17–18 May 2020. Considering the feasibility of the test parameters, the optimized parameters were adjusted so that machine forward speed was 3 km·h$^{-1}$, the rotation speed of the threshing drum was 550 r·min$^{-1}$, the rotation speed of the cleaning fan was 990 r·min$^{-1}$ and the scale sieve's opening was 35 mm. Three tests were carried out and the average value was taken as the test verification value. According to the mechanical industry standard of the People's Republic of China (JB/T 11912-2014), the loss rate and damage rate are required to be less than 5%, and the impurity rate must be less than 3%. The seed loss rate, breakage rate and impurity rate measured in the experiment were 2.35%, 0.22% and 0.51%, respectively, which were all lower than the standard.

## 5. Conclusions

(1) A combined Chinese milk vetch seed harvester was designed, and both its structure composition and working principle were described. Parameter design and simulation analysis were carried out on the key components, such as the flexible anti pod-dropping seedling-lifting header, the longitudinal rod-teeth-type threshing device, the air-sieve-type layered impurity-controlled cleaning device, etc.

(2) The optimization model of harvesting parameters of the Chinese milk vetch green manure seed harvester was established, and the multi-parameter optimization was obtained when seed loss rate, breakage rate and impurity rate were the smallest: machine forward speed was 3 km·h$^{-1}$, the rotation speed of the threshing drum was 550 r·min$^{-1}$, the rotation speed of the cleaning fan was 990 r·min$^{-1}$ and the scale sieve's opening was 35 mm. Under these parameter conditions, the field test was carried out, and results showed that the seed loss rate was 2.35%, the breakage rate was 0.22% and the impurity rate was 0.51%, which was better than the loss rate and breakage rate specified in the relevant standards, less than 5%, and the impurity rate, less than 3%.

(3) The production efficiency of the developed Chinese milk vetch seed combine harvester can reach 0.53~0.87 hm$^2$·h$^{-1}$, and it can effectively solve the shortage problem of efficient seed harvest equipment in large-scale planting areas of Chinese milk vetch. At present, only one variety of Chinese milk vetch was tested in this study, but the research will help to carry out experiments on different varieties of Chinese milk vetch and other green manure varieties in paddy fields to further verify the operational performance and adaptability of the designed Chinese milk vetch seed combine harvester.

**Author Contributions:** Conceptualization, Z.Y. and J.Y.; methodology, Z.Y. and H.W. (Hai Wei); software, Z.Y. and X.G.; validation, Z.Y. and H.W. (Huichang Wu); formal analysis, Z.Y. and H.W. (Hai Wei); investigation, Z.Y., T.H. and J.W.; resources, Z.Y. and J.W.; data curation, Z.Y.; writing—original draft preparation, Z.Y. and X.G.; writing—review and editing, Z.Y., X.G. and J.Y.; visualization, J.Y. and X.G.; supervision, H.W. (Huichang Wu); and J.W.; project administration, H.W. (Huichang Wu); funding acquisition, H.W. (Huichang Wu); All authors have read and agreed to the published version of the manuscript.

**Funding:** This research was funded by China Agriculture Research System—Green Manure, grant number CARS-22; Basic Scientific Research Service Fee of Chinese Academy of Agricultural Sciences, grant number S202012; Innovation Engineering of the Chinese Academy of Agricultural Sciences—Primary Processing Equipment for Major Grain and Economic Crops, grant number 31-NIAM-09.

**Institutional Review Board Statement:** Not applicable.

**Informed Consent Statement:** Not applicable.

**Data Availability Statement:** The data presented in this study are available on-demand from the first author at (youzhaoyan@caas.cn).

**Acknowledgments:** The authors would like to acknowledge Wuhu Qingyijiang Seed Industry Co., Ltd. for its strong support in the trial production and processing of the Chinese milk vetch seed combine harvester, and for providing the test site for this study.

**Conflicts of Interest:** The authors declare no conflict of interest.

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
