# Peer review of "Design and Multi-Parameter Optimization of a Combined Chinese Milk Vetch (Astragalus sinicus L.) Seed Harvester"

_agriculture, doi:10.3390/agriculture12122074_

Round 1

Reviewer 1 Report

·         I could not show all the units in the article, but the author should definitely correct all the units in the publication in the way I have pointed out...

·          Must be min-1

·          Mjust be km h-1

·         Line 32-33-34….. Production area and production amount of Chinese milk vetch should be written

·         Line 36….. The statement "has strong nitrogen fixation ability" should be supported by a figüre

·          Must be kg hm-2

·         Line 52-53…..” The seed impurity rate and breakage rate are high, which seriously affects the scale promotion and application of Chinese milk nut” values should be given, adding the source

·         Line 57…..” There are few reports on the mechanized seed harvesting of Chinese milk vetch.” Which report? Tehy must be given

·         Figure 1 is too complicated and the quality is low, parts 6,7 and 8 of the machine should be given in detail as separate figures

·          kg s-1

·          Must be m s-1

·          Must be hm2 h-1

·         Figure 5…quality is very bad, must be changed

·          Must be g m-2

·         Line 247….”The average moisture content of seeds was 10.2%.”

·         How was the moisture content measured? There is no information about this, it should be included in the publication.

·         Where and in what year were the trials conducted? What is the size of the land? I didn't see any information about these, should be added

·         "Chinese milk vetch plants" I would like to see a real picture of this plant in the field before it is harvested. It should be included in the publication

·         Line 275-279….. Values should be given about these selected test variables

·         Which program was used for variance analysis and what is the number of replicates? Which experimental design was used for statistical analysis?

·         How the machine feed speed and fan rotation speed were calculated. There is no information about these, they must be added....

·         Figure 8 and 9 and 10. Quality must be corrected

·         What does this publication ultimately offer us for the future?

·         How was the data used in the computer program to create the model in Figure 5 determined?

·         Are these values from the literature?

·         Did you create the model in Figure 5 or is it  from the literature?

·         How was the fan speed (min -1) found?

·         Publish the model, brand and accuracy of all measuring instruments used in the experiment.

·         What is its contribution to science? These should be given in the conclusion

Author Response

Response to Reviewer 1 Comments

Thanks very much for taking your time to review my manuscript. I really appreciate all your comments and suggestions! Please find my itemized responses in below and my revisions in the resubmitted files.

I’ll response to the questions one by one.

Point 1: I could not show all the units in the article, but the author should definitely correct all the units in the publication in the way I have pointed out...

Response 1: all the units in the article have been corrected, for example, km·h-1、r·min-1、kg·hm-2、kg·s-1、m·s-1、, etc.

Point 2: Line 32-33-34….. Production area and production amount of Chinese milk vetch should be written.

Response 2: the production area and production amount of Chinese milk vetch were written in red font in 1. Introduction.( In the 1970s, the planting area of Chinese milk vetch once exceeded 6.7 million square kilometers, mainly distributed in Sichuan, Hubei, Hunan, Jiangxi, Anhui, Jiangsu, Zhejiang and other provinces to the south of the Yangtze River.)

Point 3: The statement "has strong nitrogen fixation ability" should be supported by a figure.

Response 3: the “strong nitrogen fixation ability” has been supported by a figure, see Figure 1(a). Nitrogen-fixation effect picture of Chinese milk vetch in seedling stage.

Point 4: The seed impurity rate and breakage rate are high, which seriously affects the scale promotion and application of Chinese milk vetch” values should be given, adding the source

Response 4: although the seed impurity rate and breakage rate are high by using the existing harvest method, but actually, the specific values of impurity rate and breakage rate have not been provided by anyone or any reference, so I have deleted the sentence in the paper.

Point 5: There are few reports on the mechanized seed harvesting of Chinese milk vetch.” Which report? They must be given

Response 5: few reports means almost no reports have been found about, before the research, no reference has been found about Chinese milk vetch combine harvester.

Point 6: parts 6,7 and 8 of the machine should be given in detail as separate figures.

Response 6: detail figures have been given in the paper, see Figure 4 represents part 8 and Figure 8 represents part 6, and part 7 is just a storage of seeds.

Point 7:    The average moisture content of seeds was 10.2%.”How was the moisture content measured? There is no information about this, it should be included in the publication.

Response 7: the moisture content of harvested seed was measured by PM-8188-A moisture meter, see 4.1 Experiment conditions

Point 8: Where and in what year were the trials conducted? What is the size of the land? I didn't see any information about these, should be added

Response 8: The field harvest experiment of Chinese milk vetch was conducted in Yijiang Town, Nanling County, Wuhu City, the experiment time was from May 11 to 13, 2020. The test site is about 120 m long, and 50 m wide. see details 4.1 Experiment conditions and 4.3. Experiment scheme

Point 9: "Chinese milk vetch plants" I would like to see a real picture of this plant in the field before it is harvested. It should be included in the publication

Response 9: A real picture of Chinese milk vetch during harvest was provided in Figure 1(b).

Point 10:How the machine feed speed and fan rotation speed were calculated. There is no information about these, they must be added....

Response 10: (added in article):The adjustment methods of each parameter in the test were as follows) machine forward speed adjustment: driver adjusted the infinitely variable gear to get different for-ward speeds. The forward speed was accurately detected by the speed sensor installed on the rear wheel and displayed on the instrument panel in the cab. 2) rotation speed adjustments of the threshing drum and the centrifugal fan: The V-belt continuously variable transmission was adopted, and the moving plate of belt wheel was adjusted by hydro-static and manual operation, so as to obtain different transmission ratio and different rotation speed. 3) Scale sieve’s opening adjustment: through the position rotation of the adjusting plate installed on the sieve frame, the scale’s opening could be adjusted, and the specific scale’s opening could be measured by ruler.

Point 11: Figure 8 and 9 and 10. Quality must be corrected

Response 11: Figure 8 and 9 and 10 have been changed, Figure Quality have been corrected see Figure 9、10、11.

Point 12: How was the data used in the computer program to create the model in Figure 5 determined?Are these values from the literature?Did you create the model in Figure 5 or is it from the literature?

Response 12: Since I add a Figure, Figure 5 has become Figure 6. Figure 6 was created from software, actually I realize that the simulation results should be compared with the actual results to show whether the wind speed in different areas of the fan meets the requirements of cleaning. So the actual test verification and comparison of airflow velocity distribution in the fan have been added in this paper, see “3.3.2 Field test verification of centrifugal fan air flow” for details.

Point 13 : How was the fan speed found?

Response 13: The air velocity of each measuring point measured by AR856 digital anemometer. see “3.3.2 Field test verification of centrifugal fan air flow” for details.

Point 14 : What does this publication ultimately offer us for the future? What is its contribution to science? These should be given in the conclusion.

Response 14:See details in conclusion (3). The production efficiency of developed Chinese milk vetch seed combine harvester can reach 0.53~0.87 hm2·h-1, it can effectively solve the shortage problem of efficient seed harvest equipments in large-scale planting area of Chinese milk vetch. At present, only one variety of Chinese milk vetch was tested in this study, the research would further help to carry out experiments on different varieties of Chinese milk vetch and other green manure varieties in paddy fields.

Best wishes!

Reviewer 2 Report

The full text is clear in thinking and standardized in writing. The developed a combined Chinese milk vetch green manure seed harvester has certain innovation and has certain guidance for actual production. As far as the content of text writing is concerned, there are still some details that need to be paid attention to.

1、 At the beginning of the second paragraph of the introduction section of Section I, on what basis improvements and optimizations have been made, and whether other relevant studies are cited, please specify.

2、 In table 1, the authors should consistently write the maxium feeding quantity unit throughout the manuscript. Please, replace the maxium feeding quantity (kg.s-1) with the maxium feeding quantity (kg/s) .

3、 There should be a space between the value and the unit, please check the whole text and change it.

4、 3.3.1 section simulation part, the simulation results should be compared with the actual do, and explain whether the wind speed in different areas of the fan to meet the requirements of clear selection.

5、 The Lack of fit term in Table 5 is an important data used to assess the reliability of the equation. If it is significant, it indicates that the equation is poorly simulated and needs to be adjusted. If it is not significant, it indicates that the equation is simulated relatively well and can be analyzed well for future data. It cannot be omitted, so please add it in the corresponding place in the table.

Author Response

Thanks very much for taking your time to review my manuscript. I really appreciate all your comments and suggestions! Please find my itemized responses in below and my revisions in the resubmitted files.

I’ll response to the questions one by one.

Point 1: At the beginning of the second paragraph of the introduction section of Section I, on what basis improvements and optimizations have been made, and whether other relevant studies are cited, please specify.

Response 1: (added in article )The Chinese milk vetch(Astragalus sinicus L.)green manure seed combine harvester developed in this paper was based on “World Group 4LZ-5.0E Ryzen Grain Harvester”. The structure and movement parameters of the key components, such as the flexible anti pod-dropping seedling-lifting header, the longitudinal rod teeth type threshing device and the air-sieve type layered impurity-controlled cleaning device were newly designed and optimized according to the harvest characteristics of this green manure variety.

Additionally, during the writing of the paper, we also cited the relevant literatures, see references such as 14、20-24.

Point 2 : In table 1, the authors should consistently write the maxium feeding quantity unit throughout the manuscript. Please, replace the maxium feeding quantity (kg.s-1) with the maxium feeding quantity (kg/s) .

Response 2: The whole text has been checked and the inconsistencies of units in the paper have been revised, for example, km·h-1、r·min-1、kg·hm-2、kg·s-1、m·s-1, etc.

Point 3 : There should be a space between the value and the unit, please check the whole text and change it..

Response 3: The whole text has been checked, all values and units have been separated by spaces, for example, 0.025 kg·s-1q=5 kg·s-1、3.31 g·m-2.

Point 43.3.1 section simulation part, the simulation results should be compared with the actual do, and explain whether the wind speed in different areas of the fan to meet the requirements of clear selection.

Response 4: The actual test verification and comparison of airflow velocity distribution in the fan have been added in this paper, see “3.3.2 Field test verification of centrifugal fan air flow” for details, and it could be seen that the airflow velocity distribution of each outlet was consistent. The horizontal airflows both at the upper and lower outlets were more uniform, which were conducive to the precleaning of threshed materials and the discharge of impurities in the tail. The middle outlet presented the law of high airflow velocity in the middle and low airflow on both sides, which was conducive to the blowing and stratification of the threshed mixture during the falling process.

Point 5: The Lack of fit term in Table 5 is an important data used to assess the reliability of the equation. If it is significant, it indicates that the equation is poorly simulated and needs to be adjusted. If it is not significant, it indicates that the equation is simulated relatively well and can be analyzed well for future data. It cannot be omitted, so please add it in the corresponding place in the table.

Response 5:“Lack of fit”is supplemented in Table 5, and the values of“Lack of fit”are 0.0796、0.5761 and 0.2077 respectively,all the values are greater than 0.05, which indicate that the reliabilities of equations are good.

Best wishes!

Reviewer 3 Report

The title of the manuscript is Design and Multi-Parameter Optimization of a Combined Chinese Milk Vetch Green Manure Seed Harvester. The topic is interesting, and the research results can provide a reference for design of the Chinese milk vetch seed combine harvester. However, there are some problems in the manuscript, and the major revision is needed before published.

1. The title is hard to understand, and it should be revised.

2.The language needs major revision, especially the technical terms, such as “threshing cylinder” or “threshing drum”, should not be “threshing roller”.

3. The manuscript should focus on the innovative design of some specific mechanism and related analysis and experimental study.

4. The factors of the experiment should be more reasonable. For example, the factors for the threshing cylinder should be the speed and clearance; and the levels of the factors should be variable in design, and it should be adjustable according to the condition of crops in actual field operation, and most of the levels should not be fixed and, the result may be used to determine the range of levels of the factors under the experimental conditions of the crop.

5. What is the purpose of numerical simulation? And how does the result support the design?

6. Is the regression model lack of fit?

7. The significance of the interaction should be analyzed.

8. The control methods of test variables should be elaborated.

9. Line 326: the annotation of significance is wrong.

10. Error calculation formula should be given.

Author Response

Thanks very much for taking your time to review my manuscript. I really appreciate all your comments and suggestions! Please find my itemized responses in below and my revisions in the resubmitted files.

I’ll response to the questions one by one.

Point 1: The title is hard to understand, and it should be revised.

Response 1: The title has been changed from “Design and Multi-Parameter Optimization of a Combined Chinese Milk Vetch Green Manure Seed Harvester” to “Design and Multi-Parameter Optimization of a Combined  Chinese Milk Vetch Astragalus sinicus L.Seed Harvester

Point 2: The language needs major revision, especially the technical terms, such as “threshing cylinder” or “threshing drum”, should not be “threshing roller”.

Response 2: The inappropriate english expressions in the article have been revised, especially some technical terms, such as “threshing roller” in the paper has been changed to “threshing drum”, etc.

Point 3: The manuscript should focus on the innovative design of some specific mechanism and related analysis and experimental study.

Response 3: The innovative designs of flexible anti pod-dropping seedling-lifting header, longitudinal rod-teeth type threshing drum and air-sieve type layered impurity-controlled cleaning device have been given in the paper, see Figure 3、Figure 4 and Figure 8,and some related analysis and experimental study has been added, for example, the simulation results and actual experimental verification and comparison of airflow velocity distribution in the fan have been added in this paper, see “3.3.2 Field test verification of centrifugal fan air flow” for details.

Point 4: The factors of the experiment should be more reasonable. For example, the factors for the threshing cylinder should be the speed and clearance; and the levels of the factors should be variable in design, and it should be adjustable according to the condition of crops in actual field operation, and most of the levels should not be fixed and, the result may be used to determine the range of levels of the factors under the experimental conditions of the crop.

Response 4: Actually the factors are variable, The adjustment methods of each parameter in the test were as follows: 1) machine forward speed adjustment: driver adjusted the infinitely variable gear to get different for-ward speeds. The forward speed was accurately detected by the speed sensor installed on the rear wheel and displayed on the instrument panel in the cab. 2) rotation speed      adjustments of the threshing drum and the centrifugal fan: The V-belt continuously variable transmission was adopted, and the moving plate of belt wheel was adjusted by hydro-static and manual operation, so as to obtain different transmission ratio and different rotation speed. 3) Scale sieve’s opening adjustment: through the position rotation of the ad-justing plate installed on the sieve frame, the scale’s opening could be adjusted, and the specific scale’s opening could be measured by ruler. (This paragraph has been added to the manuscript see details in 4.3)

Point 5: What is the purpose of numerical simulation? And how does the result support the design?

Response 5:The actual test verification and comparison of airflow velocity distribution in the fan have been added in this paper, see “3.3.2 Field test verification of centrifugal fan air flow” for details, and it could be seen that the airflow velocity distribution of each outlet was consistent. The horizontal airflows both at the upper and lower outlets were more uniform, which were conducive to the pre-cleaning of threshed materials and the discharge of impurities in the tail. The middle outlet presented the law of high airflow velocity in the middle and low airflow on both sides, which was conducive to the blowing and stratification of the threshed mixture during the falling process, so that the wind speed in different areas of the fan could meet the requirements of clean selection operation.

Point 6: Is the regression model lack of fit?

Response 6:“Lack of fit”is supplemented in Table 5, and the values of“Lack of fit”are 0.0796、0.5761 and 0.2077 respectively,all the values are greater than 0.05, which indicate that the reliabilities of equations are good.

Point 7:  The significance of the interaction should be analyzed.

Response 7: The significance of the interaction has been added, see “4.4.1. Model establishment and significance verification” and “4.4.2. Effect analysis of interaction factors on harvest indexes” for details.

Point 8: The control methods of test variables should be elaborated.

Response 8: The control methods of test variables has been elaborated, see 4.3 for detail, each factor in the test is variable and controllable.

Point 9: Line 326: the annotation of significance is wrong.

Response 9: the annotation of significance has been changed as follows:

** indicates that the factors have a significant influence on the test index (0.01<P≤0.05),

*** indicates that the factors have a very significant influence on the test index (P≤0.01).

Point 10: Error calculation formula should be given.

Response 10: The ereor was caculated by the Box-Behnken design method included in the Design-Expert software, just like P value, F value, and Lack of fit in Table 6.

Best wishes!

Reviewer 4 Report

In this manuscript, a combined Chinese milk vetch green manure seed harvester was designed and the parameters of key components were optimized. Overall, the works was interesting with good structure, reliable results and full discussion. As a result, it worth of publishing in the journal of Agriculture. However, some concerns need to be fixed.

(1). Before the L55-57, more relevant detailed research about references [9-14] should be presented.

(2). The word “where” in L141 and L269 should not be indented.

Author Response

Thanks very much for taking your time to review my manuscript. I really appreciate all your comments and suggestions! Please find my itemized responses in below and my revisions in the resubmitted files.

I’ll response to the questions one by one.

Point 1: Before the L55-57, more relevant detailed research about references [9-14] should be presented.

Response 1: More relevant detailed researches have been added as follows:

At present, existing researches on seed combine harvesters both at home and abroad mainly focus on food crops such as rice, wheat, maize, as well as commercial crops such as rapeseed, soybean and flax. Wang et al. [9] designed a cutting table to be a stepless speed adjustable telescopic structure to harvest rapeseed, the threshing device was designed to be a longitudinal-axis drum with the same diameter and different speed. Wang et al. [10] developed a cleaning device with segmented vibrating screens whose holes were round so fact that the cleaning rate and loss rate of maize grain could meet the   requirements of national standards for maize grain harvester. Zhang et al. [11] used Plackett-Burman test method to study the impacts of vibration screen amplitude, crank revolving speed, fan revolving speed, and fan dip angle on cleaning loss ratio and impurity percentage of rapeseed based on a two-roller and air-screen field mobile harvest test bed. Jin et al. [12] used a series of field trials to explore the influence of nine key working parameters on the quality of soybean harvesting operations, and figured out the optimal combination of parameters systematically. Shi et al. [13] designed a track combine harvester for hilly mountain flax, which included a crawler-type walking system, a low damage header to prevent winding, a transverse-flow beater with the grain rod and rod teeth with small ta-per, narrow-grid concave plates, but the impurity rate was relatively high. And reference [14] has been deleted, because it's not very relevant.

Point 2: The word “where” in L141 and L269 should not be indented.

Response 2: The word “where” in two places of the article are not indented in the revised manuscript, see details in the submitted manuscript.

Best wishes!

Round 2

Reviewer 1 Report

The sensitivity and features of the AR856 digital anemometer should be added.

The sensitivity and characteristics of the PM-8188-A moisture meter should be added.

Author Response

Thanks very much for taking your time again to review my resubmitted manuscript. I really appreciate all your comments and suggestions!

Please find my itemized responses in below and my revisions in the resubmitted files.

1、the sensitivity and features of the digital anemometer are as follows: airflow speed measurement ranges from 0.3 m·s-1 to 45.0 m·s-1, the measurement error is  ± 3%, the resolution ratio was 0.001 m·s-1.

2、the moisture content of harvested seed was measured by PM-8188-A moisture meter, the moisture meter measurement ranges from 1% to 40%, the sample capacity is 240 ml, the temperature range is 0~40 ℃, the measurement accuracy is 0.5% at the basis of  drying method, the water content of harvested seeds was measured three times, and the average value was 10.2%. 
